# Can Color Doppler Ultrasound Be Effectively Used as the Follow-Up Modality in Patients Undergoing Splenic Artery Aneurysm Embolization? A Correlational Study between Doppler Ultrasound, Magnetic Resonance Angiography and Digital Subtraction Angiography

**DOI:** 10.3390/jcm12030792

**Published:** 2023-01-19

**Authors:** Krzysztof Lamparski, Grzegorz Procyk, Krzysztof Bartnik, Krzysztof Korzeniowski, Rafał Maciąg, Vadym Matsibora, Michał Sajdek, Alicja Dryjańska, Emilia Wnuk, Grzegorz Rosiak, Edyta Maj, Magdalena Januszewicz, Aleksandra Gąsecka, Tomasz Ostrowski, Piotr Kaszczewski, Zbigniew Gałązka, Mikołaj Wojtaszek

**Affiliations:** 12nd Department of Clinical Radiology, Medical University of Warsaw, Banacha 1a, 02-097 Warsaw, Poland; 21st Chair and Department of Cardiology, Medical University of Warsaw, Banacha 1a, 02-097 Warsaw, Poland; 3Department of General, Endocrine and Vascular Surgery, Medical University of Warsaw, Banacha 1a, 02-097 Warsaw, Poland; 4Everlight Radiology, 350 Euston Rd, London NW1 3AX, UK

**Keywords:** splenic artery aneurysm, visceral artery aneurysm, aneurysm coil embolization, color Doppler ultrasound, aneurysm follow-up, aneurysm surveillance

## Abstract

Splenic artery aneurysm (SAAs) rupture is associated with a high mortality rate. Regular surveillance with imaging before and after intervention is crucial to guide best evidence treatment. The following study aimed to determine the efficacy of color Doppler ultrasound imaging (DUS) compared to digital subtraction angiography (DSA) and magnetic resonance angiography (MRA) as a follow-up modality after selective coil embolization of true SAAs. We analyzed data from 20 patients, 15 females (48.1 ± 16.1 years) undergoing selective SAA coil embolization using detachable fibered embolization coils. Imaging using DUS, MRA, and DSA was performed 3 months after the initial embolization or the consequent re-embolization procedure. Primary clinical success, defined as Class I aneurysm occlusion, on 3-month follow-up was seen in 16 (80.0%) patients. DUS had a sensitivity of 94.4% and a specificity of 42.9% when compared to DSA and 92.3% and 30%, respectively, when compared to MRA in identifying Class I aneurysm occlusion. The positive predictive value (PPV) of DUS in identifying the need for re-embolization was 75.0%, while the NPV of DUS in these terms was 90.5%. DUS showed a high sensitivity in detecting aneurysm occlusion and clinical success, simultaneously exhibiting poor specificity. Still, with caution, this follow-up modality could be used for monitoring select low-risk patients after selective embolization of SAAs. DUS could provide a higher cost-to-benefit ratio, enabling more systematic post-procedural follow-up, as it is far more commonly used compared to MRA and non-invasive compared to DSA.

## 1. Introduction

True splenic artery aneurysms (SAAs) have an estimated incidence of 0.5–2%, representing approximately 50–60% of all visceral aneurysms [1,2]. The most serious complication is SAA rupture, which is associated with a mortality rate of 10–25% [3]. The 2005 ACC/AHA guidelines recommend treatment of true SAAs, which are at least 2.0 cm in diameter in women of childbearing age who are not pregnant, and in patients of either gender undergoing liver transplantation (Class I, Level of Evidence: B). Treatment should be also considered in the case of true SAAs at least 2.0 cm in diameter in women beyond childbearing, in men and in all false aneurysms (Class IIa, Level of Evidence: B) [4]. False aneurysms, or pseudoaneurysms, differ from true aneurysms in that they lack all three normal elements of the arterial wall, and they are usually a result of vessel trauma.

Due to its high efficacy and relatively low risk of complications, endovascular procedures have become the current mainstay of SAA treatment [5]. These can be performed non-selectively by occluding the main splenic artery, or by selectively by coil embolization of the aneurysm with preservation of the splenic artery. The latter prevents splenic ischemia and offers a lower complication rate in regard to post-embolization syndrome, splenic abscesses, or portal vein thrombosis [6]. Selective coil embolization requires periodic follow-ups to assess aneurysm growth and possible aneurysm reperfusion [7,8]. Currently, none of the guidelines address either the optimal follow-up modality or the best time interval after which follow-up imaging should be performed. Digital subtraction angiography (DSA) is considered the reference method, although recent studies have shown that magnetic resonance angiography (MRA) has a higher sensitivity in assessing aneurysm sac reperfusion [9,10]. No evidence is available on the efficacy of color Doppler ultrasound imaging (DUS) in the follow-up after selective coil SAA embolization. Therefore, we aimed to determine the potential efficacy of DUS as a follow-up modality used after selective coil embolization of SAA with preservation of the main splenic artery in relation to MRA and DUS.

## 2. Materials and Methods

### 2.1. Patient Population

In this retrospective study, we included 20 consecutive patients undergoing selective embolization of true SAAs using detachable fibered embolization coils (Concerto, Medtronic, Minneapolis, MN, USA) between March 2013 and April 2020 in our tertiary referral center. Preoperative imaging was based on either computed tomography (CT) or DUS, or both, and was used for procedure planning. Indications for treatment were based on the ACC/AHA 2005 Practice Guidelines and included: (i) minimum aneurysm size of 20 mm, (ii) aneurysm growth during observation, (iii) women of childbearing age, or (iv) patients concerned about the risks associated with receiving no treatment [4].

### 2.2. Treatment Procedure

The embolization procedure was performed by two experienced interventional radiologists, under local anesthesia, with arterial access through the common femoral artery or the left axillary artery. After vascular access was obtained, 3000 IU of heparin was administered intravenously followed by 1000 IU of heparin after each subsequent hour of surgery. In the case of wide-necked aneurysms, the parent artery was supported with a self-expandable stent (Xpert self-expandable stent, Abbott Vascular, Abbott Park, IL, USA) prior to implantation of the fibered embolization coils using the stent-assisted coiling (SAC) technique. Narrow-necked aneurysms were treated using the coil packing (CP) embolization technique. All patients received a single antiplatelet drug for the first three months after treatment or re-treatment. All procedures performed in this study involving humans were conducted in accordance with the ethical standards of the institutional and/or national research committee and with the Declaration of Helsinki. The study was officially deemed exempt from the Bioethical Medical Committee consent due to the retrospective character and no interventions associated with participation in the study.

### 2.3. Follow-Up Methods, Techniques, and Parameters

All patients underwent post-procedural follow-up imaging 3 months after the initial procedure or 3 months after a secondary redo procedure; investigations included: (i) DUS, (ii) MRA, and (iii) DSA. Two patients had contraindications to MR imaging and therefore underwent only DUS and DSA follow-up imaging. All examinations were performed during the same admission, with a maximum interval of one day between each follow-up modality. Aneurysm occlusion/reperfusion was characterized with the use of a classification system adapted from Roy et al. for the evaluation of aneurysm reperfusion after cerebral aneurysm coiling [11]. Aneurysm occlusion was classified as follows: (i) Class I—complete occlusion, (ii) Class II—residual neck, and (iii) Class III—residual aneurysm. Detailed cross-sectional image analysis and evaluation were performed using a dedicated diagnostic workstation (SyngoVia, Siemens, Erlangen, Germany). All available datasets were reviewed independently by two interventional radiologists. Inconsistent cases, in which the radiologists had disagreed, were reviewed together to reach a consensus. DUS was performed by an interventional radiologist specialized in ultrasound techniques who was blinded to the cross-sectional imaging results.

Technical success was defined as successful coil embolization of the aneurysm with retained patency of the splenic artery and without loss of perfusion within the splenic parenchyma. Clinical success was defined as Class I aneurysm occlusion at a 3-month follow-up, confirmed on both DSA and MRA imaging. Class II and III aneurysm occlusion at 3-month follow-up, which usually required a re-embolization procedure, was classified as a clinical failure.

All DUS examinations were performed on a Toshiba Aplio 500 (Toshiba Medical Europe, Zoetermeer, The Netherlands) with the use of a Convex-3.5/5.0 Mhz probe and a dedicated abdominal preset that underwent further adjustment to obtain the best possible visualization conditions. The pulse-repetition frequency was set according to the flow velocity to avoid aliasing. The color gain was set immediately below the noise threshold to maximize the likelihood of sac reperfusion visualization. The Doppler angle was set between 20° and 60°, and the values of the sampling gate were between 1.5–3 mm, depending on the vessel size. The study protocol included assessment of (i) flow in the visceral trunk origin, including peak systolic velocity (PSV) and resistance index (RI), and (ii) spleen parenchyma, volume, and perfusion by both Doppler and power Doppler. Following this, the embolized aneurysm was evaluated. Hyperechoic reflections from embolization coils were assessed in B-mode followed by the assessment of aneurysm sac reperfusion in both Doppler and power Doppler options. As abdominal obesity and intestinal meteorism are known acoustic barriers for DUS imaging of vascular lesions, all patients were examined after overnight fasting. In case of poor visualization conditions in the supine position, the patient was examined in the right lateral decubitus position to achieve the best possible image quality.

Contrast-enhanced magnetic resonance angiography (CE-MRA) was used to evaluate all patients after coil embolization. This was performed on a 1.5 Tesla MRI system (Avanto, Siemens, Medical Solutions, Erlangen, Germany) with a GeneRalized Autocalibrating Partial Parallel Acquisition (GRAPPA) technique and a body matrix coil with 12 receiving elements. For the morphological assessment of the aneurysm and adjacent structures, axial T1 vibe fat-saturated breath-hold sequences were obtained using the following parameters: repetition time (TR)—3.05 ms, echo time (TE)—1.14 ms, fip angle—10°, field-of-view (FOV)—340 mm, slice thickness—2.5 mm, slices per slab—80, resolution—288, voxel size 1.6 × 1.2 × 2.5 mm, integrated Parallel Acquisition Techniques (iPAT)—2, and acquisition time—17 s. Next, coronal breath-hold fat-saturated spoiled gradient-recalled echo sequences (fast low-angle shot—FLASH 3D) were performed before and after intravenous bolus administration of paramagnetic contrast material. The sequences were characterized by the following parameters: TR—2.96 ms, TE—1.01 ms, fip angle—30°, FOV 340 mm, slice thickness—1.5 mm, slices per slab—44, resolution—256, voxel size 1.5 × 1.3 × 1.5 mm, and iPAT—2. Prior to contrast administration, only one measurement was obtained with an acquisition time of 9 s. After contrast administration, four measurements were performed with pauses at 0 s, 5 s, and 5 s, with a total acquisition time of 46 s. A dose of 0.1 mL/kg gadobutrol (Gadovist 1 mmol/mL, Bayer, Leverkusen, Germany) was administered with an automatic injector (Solaris, Medrad, Leverkusen, Germany) at a rate of 3.0 mL/s, and flushed with 20 mL of saline solution at the same flow rate. Sequence acquisitions were followed by automatic post-processing: subtraction and maximum intensity projection (MIP) reconstruction.

Angiography was performed in at least three projections through a 4F diagnostic catheter placed in the splenic artery. After selective catheterization of the celiac trunk, followed by the catheterization of the splenic artery, anteroposterior, lateral, and oblique (±45°) images were obtained. The embolized aneurysm was visualized in all these views. A nonionic iodinated contrast agent (Iomeprol 350 mgI/mL, Iomeron 350, Bracco, Milan, Italy) was used for imaging in a dose of 12 mL with an injection rate of 2–4 mL/s for each of these views. Each examination was conducted with the following parameters: FOV—33 cm, matrix—1042 × 1042 and spatial resolution—0.32 × 0.32 mm. If Class III reperfusion was demonstrated, simultaneous implantation of additional coils was performed. In cases where Class II reperfusion was evident on both DSA and MRA, a decision on whether or not to implant additional coils was jointly made by two experienced interventional radiologists. The decision was based on the relevance of the leak and the technical feasibility of a safe re-embolization procedure. When the leak was visible only on MRA (not visible on DSA), no attempt was made to place further coils. All patients were informed and consented to a re-do embolization procedure prior to follow-up angiography.

### 2.4. Study Endpoints

The primary study endpoint was defined as the sensitivity and specificity of DUS in diagnosing Class I aneurysm occlusion (the clinical success) in relation to DSA as the current benchmark imaging modality.

The co-secondary study endpoint was the sensitivity and specificity of DUS in diagnosing Class I aneurysm occlusion (the clinical success) in relation to MRA as a method with an assumed higher sensitivity and specificity than the benchmark DSA imaging modality. Another co-secondary study endpoint was the positive predictive value (PPV) and the negative predictive value (NPV) of DUS in identifying the need for re-embolization. This was calculated as Class II/Class III finding on DUS in relation to the actually performed re-embolization.

### 2.5. Statistical Analysis

Statistical analyses were performed using SAS software (Statistical Analysis System version 9.4, SAS Institute Inc., Cary, NC, USA). Categorical variables are shown as counts and percentages. Continuous variables are shown as means ± standard deviations.

## 3. Results

### 3.1. Patients’ Characteristics

A total of 20 patients (15 women and 5 men) were enrolled in this study and 25 embolization procedures were performed. Primary coil packing was performed in 14 procedures, while primary stent-assisted coiling was performed during six procedures. Visualization conditions for DUS studies were classified as poor in five patients. The largest contributing factor was the location of the aneurysm, and this occurred in four out of five patients. Finding aneurysms located in the middle segment of the splenic artery was significantly hampered by difficulties in acquiring an adequate acoustic window. Detailed patients’ baseline characteristics and splenic artery aneurysms’ characteristics are presented in Table 1.

### 3.2. Treatment Results

Technical success was achieved in all patients with no peri-procedural complications. Primary clinical success at 3-month follow-up was observed in 16 (80.0%) patients. Three (15.0%) patients underwent re-embolization for Class II aneurysm neck reperfusion. One (5.0%) patient underwent re-embolization for Class III aneurysm reperfusion and required two additional procedures before secondary clinical success was achieved. The main splenic artery trunk occluded in two (10.0%) patients during the follow-up period, but this did not result in clinically significant splenic ischemia as there was adequately developed collateral circulation. Sample images obtained with the different investigated imaging modalities are presented in Figure 1.

### 3.3. Study Endpoints

The sensitivity of DUS in identifying Class I aneurysm occlusion (clinical success) in relation to DSA was 94.4%. On the other hand, the specificity of DUS in these terms was 42.9%. The detailed data concerning the comparison between DUS and DSA in identifying aneurysm reperfusion is summarized in Table 2.

The sensitivity of DUS in diagnosing Class I aneurysm occlusion (clinical success) in relation to MRA was 92.3%. On the other hand, the specificity of DUS in these terms was 30.0%. The detailed data concerning the comparison between DUS and MRA in identifying aneurysm reperfusion is summarized in Table 3

In our study, the PPV of DUS in identifying the need for re-embolization is 75.0% while the NPV of DUS in these terms is 90.5%. Detailed data presenting DUS’s ability to identify the need for re-embolization is summarized in Table 4.

## 4. Discussion

To the best of our knowledge, this is the first study that focused on DUS efficacy as a follow-up modality in patients treated with selective coil embolization for SAAs.

Although SAAs have been considered uncommon, they are being diagnosed with increasing frequency mainly due to the increasing use of advanced imaging techniques. According to the literature, up to 10% of SAAs rupture. About 40% of these ruptures occur during pregnancy, particularly during the third trimester [12]. Maternal mortality in case of ruptured SAA is estimated at 75% while fetal mortality can reach 95% [13,14].

Importantly, there is still a lack of consistency in guidelines regarding both treatment and follow-up. While computed tomography angiography (CTA) (especially with acquisition slices ≤ 1 mm) remains the consistent diagnostic modality of choice with high evidence levels, the evidence for the optimal surveillance modality after embolization is fairly weak. The 2017 European Society for Vascular Surgery (ESVS) Clinical Practice Guidelines on the Management of the Diseases of Mesenteric Arteries and Veins advise confirmation of the success of aneurysm occlusion or thrombosis after endovascular treatment with CTA, MRA, or DUS depending on the aneurysm location, but do not specify which method is appropriate for which embolization technique, i.e., stent-assisted coiling, coil packing, or ‘trapping’ [15]. Similarly, the even more recent 2020 Society for Vascular Surgery (SVS) Clinical Practice Guidelines on the Management of Visceral Aneurysms, recommend periodic surveillance with the use of CTA, DUS, or MRA to assess the possibility of endoleak or aneurysm reperfusion after endovascular treatment (level of recommendation: Grade 2, quality of evidence: B), but the conclusion to the evidence level is based on two studies published in 1996 and 1998 [16].

The optimal imaging method should be easily available, reproducible, non-invasive, and accurate. CTA imaging has been proposed by several authors for follow-up at 1 and 6-month intervals, but it is now believed that this modality is hampered by excessive beam hardening artifacts making the assessment of aneurysm reperfusion unreliable [7,8].

Several studies have evaluated the use of MRA in comparison to DSA in the follow-up of patients who underwent embolization of visceral aneurysms [17]. Iryo et al. reported that CE-MRA performed immediately after and 3, 6, and 12 months after the embolization procedure had a 93% agreement with DSA during follow-up at 6 months after treatment. Moreover, CE-MRA was shown to be superior in assessing small residual neck reperfusion [9]. Yasumoto et al. reported a protocol with DSA performed at 6 months, 2 years, and 3 years after the endovascular treatment and CE-MRA performed 1–3 months after initial treatment [18]. Similarly, Wojtaszek et al. demonstrated the superiority of MRA over DSA in detecting aneurysm reperfusion, and proposed a simplified protocol to facilitate early reintervention in cases of aneurysm reperfusion and primary embolization failure [10].

While CTA and MRA have been explored to some extent as valid noninvasive alternatives to DSA, the utility of DUS in the follow-up of SAA embolization has received little attention until now. Most of the published data on DUS is either available as case reports or as random follow-up data, which is not part of a structured postprocedural protocol. Tolgonay et al. in 1997, first reported the use of color Doppler for the diagnosis and follow-up of splenic vein aneurysms [19]. Pilleul et al. described a group of 14 patients with splanchnic aneurysms or pseudoaneurysms (seven splenic) treated with transcatheter embolization and followed-up within the first week color Doppler and CT, and concluded that both color Doppler and CT, used as a follow-up modality, can provide substantial postoperative information [20]. Balderi et al. also performed follow-up imaging at 1, 6, and 12 months after endovascular treatment of visceral aneurysms and pseudoaneurysms using Doppler ultrasound in 2 out of 31 patients when computed tomography angiography interpretation was difficult due to artifacts attributable to metal coils [21]. More recently, Venturini et al. performed a color Doppler ultrasound to assess the technical success of SAAs’ embolization, defined as a complete aneurysm sac exclusion. Simultaneously, the same imaging modality was used to evaluate the postprocedural perfusion of the splenic parenchyma [22].

A search of the MEDLINE, EBASE databases, and the Cochrane library, yielded no studies which focused on DUS efficacy as a follow-up modality in patients treated with selective coil embolization for SAAs. Considering that DUS is a non-invasive and universally available imaging modality, it is surprising that its sensitivity and specificity in post-embolization splanchnic aneurysm follow-up has still not been assessed, especially as the method has been validated in different clinical settings. Cantisani et al. compared the efficacy of color DUS, Contrast-Enhanced Ultrasound (CEUS), CT, and MR in detecting endoleak after endovascular abdominal aortic aneurysm repair (EVAR), and demonstrated a sensitivity and specificity of DUS of 58% and 93%, respectively, which is at a similar level to the results obtained in our study [23]. Noteworthily, the authors calculated the sensitivity and specificity inversely to our methodology (here, endoleak detection was the endpoint, while the endpoint of our study was aneurysm occlusion).

Interestingly, CEUS can also be used as a supplementary method for the assessment of patients after selective SAA aneurysm embolization with a similar sensitivity but possibly higher specificity. This method works at a second harmonic ultrasound frequency which reduces artifacts; therefore, it can be considered even when conventional ultrasound fails to provide definitive information [24,25,26]. As an example, Piscaglia et al. used conventional ultrasound and color duplex-doppler ultrasonography to assess the results of coil embolization of SAAs. As these methods were unable to provide unequivocal evaluation, the authors decided on the supplementary use of CEUS to demonstrate incomplete aneurysm occlusion [24]. Unfortunately, the respective methods’ sensitivities and specificities were not calculated, hindering a comparison to the data presented in this paper.

Doppler specific signs, such as ‘swirl flow’, the ‘yin-yang’ signs in color Doppler and ‘to-and-fro’ signs in pulsed-wave Doppler, which are prevalent for pseudoaneurysms [27] were not identified here due to the small ‘leak/reperfusion’ size within the remnant aneurysm sac, and the deep location of true SAAs. Their precise assessment may be unfeasible and, therefore, was omitted in this study.

## 5. Limitations

Importantly, our study has several limitations. The main limiting factor is the small cohort size mainly due to the low SAAs prevalence. The other limitation is its single-center design—recruiting patients from many hospitals in future studies could increase the power of the results. Finally, there is a lack of long-term follow-up in our study, which precludes drawing long-range conclusions.

## 6. Conclusions and Future Perspectives

In the presented study, DUS showed a high sensitivity in detecting Class I aneurysm occlusion (clinical success), simultaneously exhibiting poor specificity in these terms. Due to its rather low efficacy in detecting small leaks (Class II), and the presence of intestinal artifacts that may prevent reliable assessment, DUS should be used with caution for monitoring patients after selective embolization of SAAs. With the understanding of this limitation, DUS in certain clinical scenarios may reduce the amount of DSA and MRA performed. MRA and DUS could be performed 3 months after surgery. If complete aneurysm occlusion was achieved (Class I) and confirmed in both modalities in the presence of good DUS imaging conditions, further follow-ups after 1, 2, and 3 years could be performed using only DUS and possibly supplemented by CEUS in complex cases. If MRA showed Class II aneurysm occlusion or technical conditions prevented reliable evaluation by DUS, MRA should remain the method of choice for further follow-up imaging. Undoubtedly, this would require further confirmation in preferably a larger group of patients, as a small patient group is a significant limitation of the presented study. Nevertheless, we believe that this could not only provide a higher cost-to-benefit ratio but enable a more systematic follow-up, as DUS is far more commonly used compared to MRA and non-invasive compared to DSA. Moreover, DUS would be especially useful in scenarios where other methods are contraindicated, e.g., in the case of patients with impaired renal function or during pregnancy and breastfeeding.

## Figures and Tables

**Figure 1 jcm-12-00792-f001:**
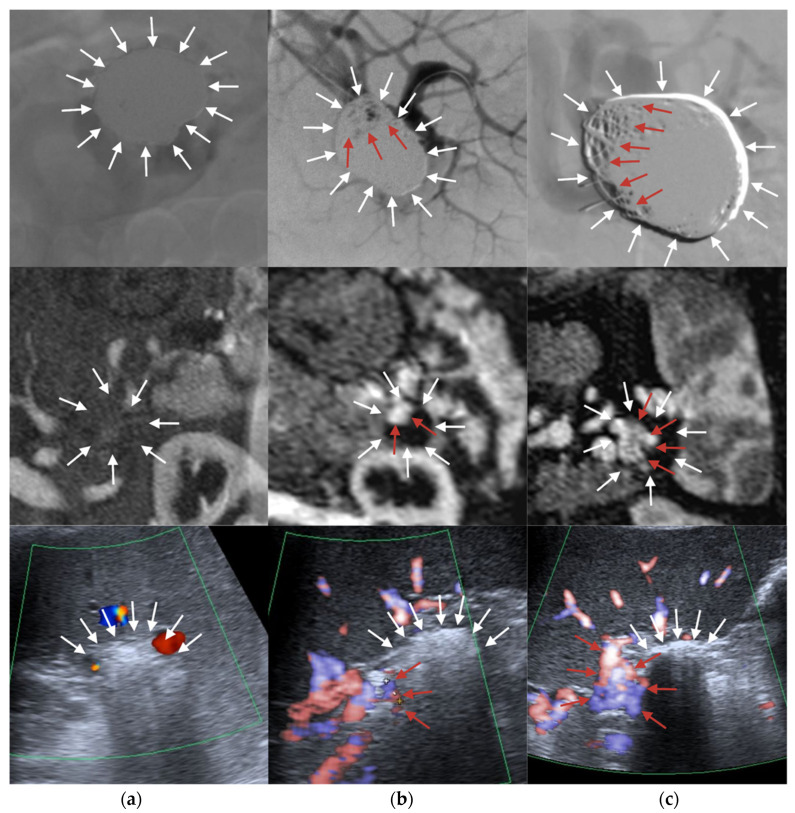
Images were taken with the use of the investigated imaging modalities: digital subtraction angiography (DSA) in the first row, magnetic resonance angiography (MRA) in the second row, and color Doppler ultrasound (DUS) in the last row. In columns, different aneurysm occlusion classes were presented, respectively, for each imaging modality: (**a**) Class I—complete occlusion; (**b**) Class II—residual neck; (**c**) Class III—residual aneurysm. White arrows—aneurysm after embolization; red arrows—aneurysm sac reperfusion.

**Table 1 jcm-12-00792-t001:** Detailed patients’ and splenic artery aneurysms’ characteristics.

Variable		Value
Sex:	Female (*n*, %)	15 (75.0%)
	Male (*n*, %)	5 (25.0%)
Age, years (mean ± SD)		48.1 ± 16.1
Aneurysm largest diameter, mm (mean ± SD)		18.1 ± 5.7
Aneurysm location:	Proximate (*n*, %)	1 (5.0%)
	Middle (*n*, %)	7 (35.0%)
	Distal (*n*, %)	12 (60.0%)
Aneurysm axial shape:	Ellipsoid (*n*, %)	14 (70.0%)
	Bilobed (*n*, %)	2 (10.0%)
	Spherical (*n*, %)	4 (20.0%)
Visualization conditions (DUS):	Good (*n*, %)	15 (75.0%)
	Poor (*n*, %)	5 (25.0%)
*n* of coils used (median, range)		10 (4–20)

SD—standard deviation; *n*—number.

**Table 2 jcm-12-00792-t002:** Cross modality comparison between DUS and DSA in diagnosing aneurysm reperfusion.

		DSA	
		Class I	Class II/Class III
DUS	Class I	17	4
	Class II/Class III	1	3

Class I—complete occlusion; Class II—residual neck; Class III—residual aneurysm; DSA—digital subtraction angiography; DUS—Doppler ultrasound.

**Table 3 jcm-12-00792-t003:** Cross modality comparison between DUS and MRA in diagnosing aneurysm reperfusion.

		MRA	
		Class I	Class II/Class III
DUS	Class I	12	7
	Class II/Class III	1	3

Class I—complete occlusion; Class II—residual neck; Class III—residual aneurysm; DUS—Doppler ultrasound; MRA—magnetic resonance angiography.

**Table 4 jcm-12-00792-t004:** DUS’s ability to diagnose the need for re-embolization.

		Re-Embolization	
		Yes	No
DUS	Class II/Class III	3	1
	Class I	2	19

## Data Availability

The data presented in this study are available on request from the corresponding author. The data are not publicly available due to privacy policy.

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
