# Peer review of "Can Color Doppler Ultrasound Be Effectively Used as the Follow-Up Modality in Patients Undergoing Splenic Artery Aneurysm Embolization? A Correlational Study between Doppler Ultrasound, Magnetic Resonance Angiography and Digital Subtraction Angiography"

_jcm, 2023, doi:10.3390/jcm12030792_

Round 1
Reviewer 1 Report
Authors reported an original article to determine the efficacy of color Doppler ultrasound imaging (DUS), compared to digital subtraction angiography (DSA) and magnetic resonance angiography (MRA) as a follow-up modality after selective coil embolization of splenic artery aneurysm (SAAs).
General considerations:
The work is interesting, the paper is very well-written and there are only a few articles in literature about this topic. I congratulate the authors for being able to recruit such a large number of patients, considering the low prevalence of SAAs.
Abstract: the abstract appropriately summarize the manuscript without discrepancies between the abstract and the remainder of the manuscript.
Keywords: inadequate. Please, reach 5 keywords.
Reference: please, follow my suggestions.
Paper
On some aspects, the authors should address:
1)In the introduction paragraph, it is necessary to clarify the distinction between aneurysm and pseudoaneurysm for beginners.
2)Could you be more precise about the technical parameters used for the Doppler US? Did you use presets or was the setting manual? What was the PRF used? You have also to specify the angle insonation, the wall filter, etc.
3)I deem it necessary that the Doppler semeiological findings of aneurysms/pseudoaneurysms be examined in the discussion. I believe that these measures can attract a greater number of readers. I suggest you read and cite the following article, in which you can find all the information that needs to be added (i.e., swirl flow, yin-yang sign, to-and-fro sign) as well as the Doppler parameters to optimize a Doppler ultrasound exam.
-Usefulness of doppler techniques in the diagnosis of peripheral iatrogenic pseudoaneurysms secondary to minimally invasive interventional and surgical procedures: imaging findings and diagnostic performance study. J Ultrasound. 2020 Dec;23(4):563-573. doi: 10.1007/s40477-020-00475-6. Epub 2020 May 20. PMID: 32436181; PMCID: PMC7588580.
4)If you deem it appropriate, I would advise you to add some concepts of CT-angiography as well.
5)Have you thought about using the contrast-enhanced ultrasound (CEUS)? I suggest you to discuss it by reading and citing the following article:
-Added value of contrast-enhanced ultrasound (CEUS) with Sonovue® in the diagnosis of inferior epigastric artery pseudoaneurysm: report of a case and review of literature. J Ultrasound. 2019 Dec;22(4):485-489. doi: 10.1007/s40477-019-00398-x. Epub 2019 Jul 20. PMID: 31327113; PMCID: PMC6838239.
6)Since the major limitation in the Doppler study of vascular lesions located in the abdomen is the acoustic barrier due to obesity and intestinal meteorism, discuss this further.
Figures:
-You could increase the iconography by adding another SAA case evaluated with Doppler US, angiography and magnetic MRA images.
Reviewer 2 Report
Authors present Can color Doppler ultrasound be effectively used as the follow-up modality in patients undergoing splenic artery aneurysm embolization? A correlational study between Doppler ultrasound, magnetic resonance angiography and digital subtraction angiography. There are some concerns in the article as follows. First, residual flow (or recanalization) in aneurysms (Class II and Class III) was not detected well by DUS compared to DSA (the specificity was 42.9 %) and MRA (the specificity was 30.0 %). Considering the poor rate, DUS is not suitable for follow-up imaging after SAA embolization. I mean you should emphasize the limitations of DUS, conversely. Also, residual flow (or recanalization) in aneurysms could have happened even after 3 months. DUS has still a risk of not detecting recanalization in the further follow-up (1, 2, and 3 years). Second, did you follow the coil compaction on the X-ray? I guess it is cost-effective for detecting the recanalization of aneurysms indirectly, and it is enough for regular follow-up after SAA embolization. If you detect coil compaction, further examination such as DSA and MRA may be required. Third, you demonstrated poor visualization of DUS in 5 cases. Was DUS visualization affected by stent-assist coiling? If this is true, this factor is also considered a limitation in DUS follow-up.
Round 2
Reviewer 2 Report
Thank you very much for the authors' effort to revise the manuscript. I have only one additional comment. Please check my comment below.
Page 4, line 167
“.. with an injection rate of 24 ml/s for…”
Do you mean the arteriography in the splenic artery was performed with an injection rate of 24 ml/s? Please clarify.
